# A Question Answering Framework for Decontextualizing User-facing Snippets from Scientific Documents

**Benjamin Newman**♡♠    **Luca Soldaini**♠    **Raymond Fok**♡
**Arman Cohan**♠◇    **Kyle Lo**♠

♠Allen Institute for AI    ♡University of Washington    ◇Yale University
{benjaminn, rayfok}@cs.washington.edu    {lucas,armanc,kylel}@allenai.org

## Abstract

Many real-world applications (e.g., note tak­ing, search) require extracting a sentence or paragraph from a document and showing that snippet to a human outside of the source doc­ument. Yet, users may find snippets difficult to understand as they lack context from the original document. In this work, we use lan­guage models to rewrite snippets from scien­tific documents to be read on their own. First, we define the requirements and challenges for this *user-facing decontextualization* task, such as clarifying where edits occur and handling references to other documents. Second, we pro­pose a framework that decomposes the task into three stages: question generation, question an­swering, and rewriting. Using this framework, we collect gold decontextualizations from ex­perienced scientific article readers. We then conduct a range of experiments across state-of-­the-art commercial and open-source language models to identify how to best provide missing-­but-relevant information to models for our task. Finally, we develop QADECONTEXT, a simple prompting strategy inspired by our framework that improves over end-to-end prompting. We conclude with analysis that finds, while rewrit­ing is easy, question generation and answering remain challenging for today's models.

⭕ github.com/bnewm0609/qa-decontext

## 1 Introduction

Tools to support research activities often rely on extracting text snippets from long, technical doc­uments and showing them to users. For example, snippets can help readers efficiently understand documents (August et al., 2023; Fok et al., 2023b) or scaffold exploration of document collections (e.g. conducting literature review) (Kang et al., 2022; Palani et al., 2023). As more applications use language models, developers use extracted snip­pets to protect against generated inaccuracies; snip­pets can help users verify model-generated out-

Figure 1: Illustration of two user-facing scenarios requir­ing snippet decontextualization. (Top) A citation graph explorer surfacing citation context snippets to explain relationships between papers. (Bottom) An AI research assistant providing snippets as attributions. Highlighted spans are added during decontextualization.

puts (Bohnet et al., 2022) and provide a means for user error recovery.

However, extracted snippets are not meant to be read outside their original document: they may include terms that were defined earlier, contain anaphora whose antecedents lie in previous para­graphs, and generally lack context that is needed for comprehension. At best, these issues make ex­tracted snippets difficult to read, and at worst, they render the snippets misleading outside their orig­inal context (Lin et al., 2003; Cohan et al., 2015; Cohan and Goharian, 2017; Zhang et al., 2023).

In this work, we consider the potential for mak­ing extracted snippets more readily-understood in user-facing settings through *decontextualiza­tion* (Choi et al., 2021)—the task of rewriting snip­pets to incorporate information from their originat­ing contexts, thereby making them "stand alone".

We focus our attention on scenarios in which users read snippets from technical documents (e.g., scientific articles). For example, consider a citation graph explorer that allows users to preview citation contexts to explain the relationship between papers (Luu et al., 2021). Also, consider an AI research assistant that surfaces extracted attribution snippets alongside generated answers. Figure 1 illustrates these two motivating applications. How do language models fare when performing snippet decontextualization over complex scientific text? Our contributions are:

First, we introduce requirements that extend prior decontextualization work (Choi et al., 2021) to handle user-facing scenarios (e.g., delineation of model-generated edits). We characterize additional challenges posed by decontextualizing scientific documents (e.g., longer text, citations and references) and describe methods to address them (§2).

Second, we propose a framework for snippet decontextualization that decomposes the task into three stages: question generation, question answering, and rewriting (§3). This decomposition is motivated by a formative study in which our framework makes decontextualization less challenging and creates higher-quality annotations. We use this framework to collect gold decontextualization data from experienced readers of scientific articles (§4).

Finally, with this data, we operationalize our framework by implementing QADECONTEXT, a strategy for snippet decontextualization (§5). Our best experimental configuration demonstrates a $41.7\%$ relative improvement over end-to-end model prompting (§5.2). We find that state-of-the-art language models perform poorly on our task, indicating significant opportunity for further NLP research. We perform extensive analysis to identify task bottlenecks to guide future investigation (§6).

## 2 Decontextualization for User-facing Snippets from Scientific Documents

In this section, we define decontextualization and motivate some additional task requirements when considering user-facing scenarios. Then, we describe additional task challenges that arise when operating on scientific documents.

### 2.1 Requirements for User-facing Snippets

**Task Definition.** As introduced in Choi et al. (2021), decontextualization is defined as:

> Given a snippet-context pair $(s, c)$, an

edited snippet $s'$ is a valid decontextualization of $s$ if $s'$ is interpretable without any additional context, and $s'$ preserves the truth-conditional meaning of $s$ in $c$.

where the context $c$ is a representation of the source document, such as the full text of a scientific article.

**Multi-sentence Passages.** While Choi et al. (2021) restrict the scope of their work to single-sentence snippets, they recommend future work on longer snippets. Indeed, real-world applications should be equipped to handle multi-sentence snippets as they are ubiquitous in the datasets used to develop such systems. For example, 41% of evidence snippets in Dasigi et al.'s (2021) dataset and 17% of citation contexts in Lauscher et al.'s (2022) dataset are longer than a single sentence. To constrain the scope of valid decontextualizations, we preserve (1) the same number of sentences in the snippet and (2) each constituent sentence's core informational content and discourse role within the larger snippet before and after editing.

**Transparency of Edits.** Prior work did not require that decontextualization edits were transparent. We argue that the clear delineation of machine-edited versus original text is a requirement in user-facing scenarios such as ours. Users must be able to determine the provenance (Han et al., 2022) and authenticity (Gehrmann et al., 2019; Verma et al., 2023) of statements they read, especially in the context of scientific research, and prior work has shown that humans have difficulty identifying machine-generated text (Clark et al., 2021). In this work, we require the final decontextualized snippet $s'$ to make transparent to users what text came from the original snippet $s$ and what text was added, removed, or modified. We ask tools for decontextualization to follow well-established guidelines in writing around how to modify quotations[1]. Such guidelines include using square brackets ([]) to denote resolved coreferences or newly incorporated information.

### 2.2 Challenges in Scientific Documents

We characterize challenges for decontextualization that arise when working with scientific papers.

**Long, Complex Documents.** We present quantitative and qualitative evidence of task difficulty compared to prior work on Wikipedia snippets.

---

[1]APA style guide: https://apastyle.apa.org/style-grammar-guidelines/citations/quotations/changes

First, Choi et al. (2021) found between 80-90% of the Wikipedia sentences can be decontextualized using only the paragraph with the snippet, and section and article titles. However, we find in our data collection (§4) that only 20% of snippets from scientific articles can be decontextualized with this information alone (and still only 50% when also including the abstract; see Table 5).

Second, we conduct a formative study with five computer science researchers, asking them to manually decontextualize snippets taken from Wikipedia and scientific papers.[2] Participants took between 30-160 seconds ($\mu$=88) for Wikipedia sentences from Choi et al. (2021) and between 220-390 seconds ($\mu$=299) for scientific snippets from our work.[3] In qualitative feedback, all participants expressed the ease of decontextualizing Wikipedia snippets. For scientific paper snippets, all participants verbally expressed difficulty of the task despite familiarity with the subject material; 3/5 participants began taking notes to keep track of relevant information; 4/5 participants felt they had to read the paper title, abstract and introduction before approaching the snippet; and 4/5 participants encountered cases of *chaining* in which the paper context relevant to an unfamiliar entity contained other unfamiliar entities that required further resolving. None of these challenges arose for Wikipedia snippets.

**Within and Cross-Document References.** Technical documents contain references to within-document artifacts (e.g., figures, tables, sections) and to other documents (e.g., web pages, cited works). Within-document references are typically to tables, figures, or entire sections, which are difficult to properly incorporate into a rewritten snippet without changing it substantially. With cross-document references, there is no single best way to handle these when performing decontextualization; in fact, the ideal decontextualization is likely more dependent on the specific user-facing application's design rather than on intrinsic qualities of the snippet. For example, consider interacting with an AI research assistant that provides *extracted* snippets:

👤 *What corpus did Bansal et al. use?*
🖥 *"We test our system on the CALLHOME Spanish-English speech translation corpus [42] (§3)."*

One method of decontextualization can be:

🖥 *"[Bansal et al., 2017] test [their] system on the CALLHOME Spanish-English speech translation corpus [42] ["Improved speech-to-text translation with the Fisher and Callhome Spanish-English speech translation corpus" at IWSLT 2013] (§3)."*

incorporating the title of cited paper *"[42]"*.[4] But in the case of a citation graph explorer, a typical interface likely already surfaces the titles of both citing and cited papers (recall Figure 1), in which case the addition of a title isn't useful. Possibly preferred is an alternative decontextualization that describes the dataset:

🖥 *"[Bansal et al., 2017] test [their] system on the CALLHOME Spanish-English speech translation corpus [42] [, a noisy multi-speaker corpus of telephone calls in a variety of Spanish dialects] (§3)."*

## 2.3 Addressing Challenges

To address the increased task difficulty that comes with working with long, complex scientific documents, we introduce a framework in (§3) and describe how it helps humans tackling this task manually. We also opt to remove all references to in-document tables and figures from snippets, and leave handling them to future work[5].

Finally, to handle cross-document references, we assume in the AI research assistant application setting that a user would have access to basic information about the current document of interest but no knowledge about any referenced documents that may appear in the snippet text. Similarly, we assume in the citation context preview setting, that a user would have access to basic information about

---

[2]Participants were all researchers with at least five published research papers at an NLP, ML or HCI venue. Four had completed PhDs while one was in a PhD program, all in computer science. All were familiar with the decontextualization task, but unfamiliar with the goals of our study.

[3]Each participant saw at least two snippets. To control for learning effects, we randomized assignment order of Wikipedia or Science.

[4]While numeric citations benefit substantially from interface assistance in surfacing information about the cited paper (Chang et al., 2023a), researchers may be able to recall details of cited papers from name-year citations like "(Post et al., 2013)". We define our target decontextualization behavior under the least-informative citation format.

[5]Real-world systems currently only surface scientific paper snippets in text-only interfaces, though one can imagine future multimodal interfaces might have different task requirements for snippet decontextualization.

the current (citing, cited) document pair but no knowledge about any other referenced documents that may appear in the snippet text.

## 3 QA for Decontextualization

Decontextualization requires resolving *what* additional information a person would like to be incorporated and *how* such information should be incorporated when rewriting (Choi et al., 2021). If we view *"what"* as addressed in our guidelines (§2), then we address *"how"* through this proposal:

### 3.1 Our Proposed Framework

We decompose decontextualization into three steps:

1. *Question generation.* Ask clarifying questions about the snippet.
2. *Question answering.* For each question, find an answer (and supporting evidence) within the source document.
3. *Rewriting.* Rewrite the snippet by incorporating information from these QA pairs.

We present arguments in favor of this framework:

**QA and Discourse.** Questions and answers are a natural articulation of the requisite context that extracted snippets lack. The relationship between questions and discourse relations between document passages can be traced to Questions Under Discussion (QUD) (Onea, 2016; Velleman and Beaver, 2016; De Kuthy et al., 2018; Riester, 2019). Recent work has leveraged this idea to curate datasets for discourse coherence (Ko et al., 2020, 2022). We view decontextualization as a task that aims to recover missing discourse information through the resolution of question-answer pairs that connect portions of the snippet to the source document.

**Improved Annotation.** In our formative study (§2.2), we also presented participants with two different annotation guidelines. Both defined decontextualization, but one (**QA**) described the stages of question generation and question answering as prerequisite before rewriting the snippet, while the other (**NoQA**) showed before-and-after examples of snippets. All participants tried both guidelines; we randomized assignment order to control for learning effects.

While we find adhering to the framework slows down annotation and does not impact annotation quality in the Wikipedia setting (§A.4), adhering

to the framework results in higher-quality annotations in the scientific document setting. 3/5 of participants who were assigned **QA** first said that they preferred to follow the framework even in the **NoQA** setting[6]. Two of them additionally noted this framework is similar to their existing note-taking practices. The remaining 2/5 of participants who were assigned **NoQA** first struggled initially; both left their snippets with unresolved acronyms or coreferences. When asked why they left them as-is, they both expressed that they lost track of all the aspects that needed decontextualization. These annotation issues disappeared after these participants transitioned to the **QA** setting. Overall, all participants agreed the framework was sensible to follow for scientific documents.

## 4 Data Collection

Following the results of our formative study, we implemented an annotation protocol to collect decontextualized snippets from scientific documents.

### 4.1 Sources of Snippets

We choose two English-language datasets of scientific documents as our source of snippets, one for each motivating application setting (Figure 1):

**Citation Graph Explorer.** We obtain citation context snippets used in a citation graph explorer from scientific papers in S2ORC (Lo et al., 2020). We restrict to contexts containing a single citation mention to simplify the annotation task, though we note that prior work has pointed out the prevalence of contexts containing multiple citations[7] (Lauscher et al., 2022).

**AI Research Assistant.** We use QASPER (Dasigi et al., 2021), a dataset for scientific document understanding that includes QA pairs along with document-grounded attributions—extracted passages that support a given answer. We use these supporting passages as user-facing snippets that require decontextualization.

### 4.2 Annotation Process

Following our proposed framework:

---

[6]For these participants, their main complaint was the act of writing down questions and answers explicitly seemed slow, but they agreed with the framework overall as intuitive.

[7]Future work could investigate decontextualization amid multiple outward references in the same snippet.

**Writing Questions.** Given a snippet, we ask annotators to write questions that clarify or seek additional information needed to fully understand the snippet. Given the complexity of the annotation task we used Upwork[8] to hire four domain experts with experience reading scientific articles. Annotators were paid $20 USD per hour[9].

**Answering Questions.** We hired a separate set of annotators to answer questions from the previous stage using the source document(s). We additionally asked annotators to mark what evidence from the source document(s) supports their answer. We used the Prolific[10] annotation platform as a high-quality source for a larger number of annotators. Annotators were recruited from the US and UK and were paid $17 USD per hour. To ensure data quality, we manually filtered a total of 719 initial answers down to 487 by eliminating ones that answered the question incorrectly or found that the question could not be answered using the information in the paper(s) (taking ∼20 hours).

**Rewriting Snippets.** Given the original snippet and all QA pairs, we ask another set of annotators from Prolific to rewrite the snippet incorporating all information in the QA pairs.

## 4.3 Dataset Statistics

In total, we obtained 289 snippets (avg. 44.2 tokens long), 487 questions (avg. 7.8 tokens long), and 487 answers (avg. 20.7 tokens long). On average, the snippets from the **Citation Graph Explorer** set have 1.9 questions per snippet while the **AI Research Assistant** snippets have 1.3 questions per snippet. Questions were approximately evenly split between seeking definitions of terms, resolving coreferences, and generally seeking more context to feel informed. See §A.2 for a breakdown of question types asked by annotators.

## 5 Experimenting with LLMs for Decontextualization

We study the extent to which current LLMs can perform scientific decontextulaization, and how our QA framework might inform design of methods.

### 5.1 Is end-to-end LLM prompting sufficient?

Naively, one can approach this task by prompting a commercially-available language model with the instructions for the task, the snippet, and the entire contents of the source paper. We experiment with ⑨ text-davinci-003 and ⑨ gpt-4-0314. For ⑨ gpt-4, most papers entirely fit in the context window (for a small number of papers, we truncate them to fit). For ⑨ davinci, we represent the paper with the title, abstract, the paragraph containing the snippet, and the section header of section containing the snippet (if available). This choice was inspired by Choi et al.'s (2021) use of analogous information for decontextualizing Wikipedia text, and we empirically validated this configuration in our setting as well (see §A.3). We provide our prompts for both models in §A.6.4 and §A.6.5.

For automated evaluation, we follow Choi et al. (2021) and use SARI (Xu et al., 2016). Originally developed for text simplification, SARI is suitably repurposed for decontextualization as it computes the F1 score between unigram edits to the snippet performed by the gold reference versus edits performed by the model. As we are interested in whether the systems *add* the right clarifying information during decontextualization, we report SARI-add as our performance metric. We additionally report BERTScore (Zhang et al., 2020) which captures semantic similarity between gold reference and model prediction, though it is only used as a diagnostic tool and does not inform our evaluative decisions; due to the nature of the task, as long as model generations are reasonable, BERTScore will be high due to significant overlap between the source snippet, prediction and gold reference.

We report these results in Table 1. **Overall, we find that naively prompting LLMs end-to-end performs poorly on this task.**

### 5.2 Can our QA framework inform an improved prompting strategy?

To improve upon end-to-end prompting, we implement QADECONTEXT, a strategy for snippet decontextualization inspired by our framework. This approach is easy to adopt, making use of widely-available LLMs as well as off-the-shelf passage retrieval models. See Figure 2 for a schematic. All prompts for each component are in §A.6.

**Question Generation.** We prompt an LLM (⑨ davinci) to generate questions with a one-shot prompt with instructions. We found more in-

---

[8] https://www.upwork.com
[9] We use the minimum wage in Washington, DC, USA ($16.10 USD at the time of annotation) as reference for determining fair compensation.
[10] https://www.prolific.co

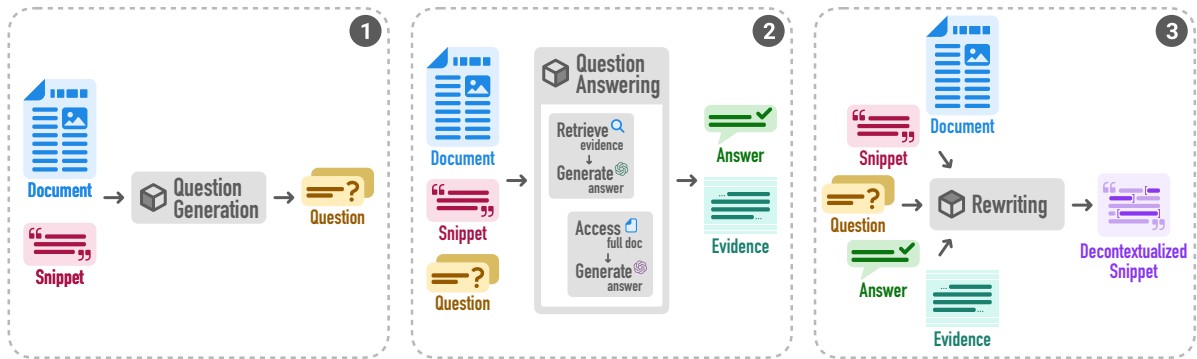

Figure 2: The three modules used for QADECONTEXT. **Question generation** ❶ formulates clarification questions given a snippet and (optionally) the source document. **Question answering** ❷ returns an answer and (optionally) supporting evidence for a given a question, snippet, and (optionally) the source document. **Rewriting** ❸ receives the snippet and (one of more elements in) the context produced by previous modules to perform decontextualization. For examples of the outputs of these steps, see Table 8.

| | **Models** | | | **Metrics** | |
|---|---|---|---|---|---|
| **Strategy** | QG | QA | R | SARI-add | BERTScore |
| QADECONTEXT | 🐦 | 🐦 | 🐦 | **0.140** | 0.483 |
| QADECONTEXT | 🐦 | 🐦 | 🐦 | **0.146** | 0.472 |
| End-to-end | | 🐦 only | | 0.135 | 0.499 |
| End-to-end | | 🐦 only | | 0.103 | 0.536 |

Table 1: Comparison between our QADECONTEXT strategy versus prompting the model end-to-end. The 🐦 davinci and 🐦 gpt-4 icons represent the models used for each of the Question Generation (QG), Question Answering (QA), Rewriting (R) components of our strategy; end-to-end prompting only uses a single model. Results from higher performance strategy are **bold**.

| **Input** | **Rewriter Module** (SARI-add) | | | |
|---|---|---|---|---|
| D Q A E | 🅰 claude | 🐦 davinci | 🐫 tülu | 🦙 llama2 |
| ✓ ✗ ✗ ✗ | 0.120 | 0.135 | 0.048 | 0.077 |
| ✓ ✓ ✗ ✗ | 0.167 | 0.198 | 0.087 | 0.090 |
| ✓ ✓ ✓ ✗ | 0.418 | 0.413 | 0.090 | 0.238 |
| ✗ ✗ ✗ ✓ | 0.142 | 0.177 | 0.069 | 0.073 |
| ✗ ✓ ✗ ✓ | 0.216 | 0.217 | 0.107 | 0.173 |
| ✗ ✓ ✓ ✓ | **0.433** | 0.422 | **0.130** | **0.330** |
| ✓ ✗ ✗ ✓ | 0.144 | 0.174 | — | 0.069 |
| ✓ ✓ ✗ ✓ | 0.199 | 0.224 | — | 0.101 |
| ✓ ✓ ✓ ✓ | 0.378 | **0.427** | — | 0.205 |
| ✗ ✓ ✗ ✗ | 0.095 | 0.097 | 0.042 | 0.041 |
| ✗ ✓ ✓ ✗ | **0.547** | **0.527** | **0.252** | **0.312** |

Table 2: Oracle performance of QADECONTEXT when using **gold** (**Q**)uestions, (**A**)nswers, answer (**E**)vidence obtained from annotators or source (**D**)ocument. Across models, extra input hurts performance (e.g., QA outperforms DQA and DQAE). Results from best two input configs are **bold**. Entries for 🐫 tülu are missing as inputs don't fit in context window.

context examples allowed for better control of the number of questions, but decreased their quality.

**Question Answering.** Given a question, we can approach answering in two ways. In *retrieve-then-answer*, we first retrieve the top $k$ relevant paragraphs from the union of the source document and any document cited in the snippet, and then use an LLM to obtain a concise answer from these $k$ paragraphs. Specifically, we use $k = 3$ and Contriever (Izacard et al., 2021) for the retrieval step, and 🐦 davinci or 🐦 gpt-4 as the LLM.

Alternatively, in the *full document* setting, we directly prompt an LLM that supports longer context windows (🐦 gpt-4) to answer the question given the entire source document as input. This avoids the introduction of potential errors from performing within-document passage retrieval.

**Rewriting.** Finally, we prompt an LLM (🐦 davinci) with the snippet, generated questions, generated answers, and any relevant context (e.g., retrieved evidence snippets if using *retrieve-then-answer* and/or text from the source document) obtained from the previous modules. This module is similar to end-to-end prompting of LLMs from §5.1 but prompts are slightly modified to accommodate output from previous steps.

**Results.** We report results also in Table 1. We find our QADECONTEXT strategy achieves **a 41.7% relative improvement** over the 🐦 gpt-4 end-to-end baseline, but given the low SARI-add scores, there remains much room for improvement.

## 5.3 Human Evaluation

We conduct a small-scale human evaluation ($n = 60$ samples) comparing decontextualized snippets with our best end-to-end (🉐 davinci) and QADE-CONTEXT approaches. Snippets were evaluated on whether they clarified the points that the reader needed help understanding. System outputs for a given snippet were presented in randomized order and ranked from best to worst. The evaluation was performed by two coauthors who were familiar with the task, but not how systems were implemented. The coauthors annotated 30 of the same snippets, and achieved a binary agreement of 70%. This is quite high given the challenging and subjective nature of the task; Choi et al. (2021) report agreements of 80% for snippets from Wikipedia.

Our QADECONTEXT strategy produces convincing decontextualized snippets in 38% of cases against 33% for the end-to-end approach. We note that decontexualization remains somewhat subjective (Choi et al., 2021), with only 42% of the gold decontextualizations judged acceptable. We conduct a two-sample Binomial test and find that the difference between the two results is not statistically significant ($p = 0.57$). See Table 4 for qualitative examples of QADECONTEXT errors.

## 6 Analyzing Performance Bottlenecks through QADECONTEXT

Modularity of our framework for decontextualization allows us to study performance bottlenecks: *Which subtask (question generation, question answering, rewriting) do LLMs struggle with the most?* We conduct ablation experiments to better understand the performance and errors of each module in QADECONTEXT. We refer the reader to Table 4 for qualitative error examples.

### 6.1 Is rewriting the performance bottleneck?

To study if the rewriting module is the bottleneck, we run *oracle* experiments to provide an upper bound on the performance of our strategy. We perform these experiments assuming that the LLM-based *rewriting* module receives *gold* (human-annotated) **Q**uestions, **A**nswers, and answer **E**vidence paragraphs. We also investigate various combinations of this gold data with the source **D**ocument itself (i.e., title, abstract, paragraph containing the snippet, and section header). To ensure our best configuration applies generally across models, we study all com-

binations using two commercial (🅰 claude-v1, 🉐 text-davinci-003) and two open source (🐫 tülu-30b, ♾ llama2-chat-70b) models. Our prompts are in §A.6.

We report results in Table 2. First, we observe that, on average, the **performance ranking of different input configurations to the rewriter is consistent across models**: (1) Including the gold evidence (E) is better than including larger document context (D), (2) including the gold answer (A) results in the largest improvement in all settings, and (3) performance is often best when the rewriter receives *only* the questions (Q) and answers (A).

Second, we find that overall performance of the best oracle configuration of QADECONTEXT (🉐 davinci) achieves 261% higher performance over the best QADECONTEXT result in Table 1. As we did not change the rewriter for these oracle experiments, **we conclude significant errors are being introduced in the question generation and answering modules, rather than in the rewriter**.

### 6.2 Are question generation or question answering the performance bottleneck?

We continue this investigation using similar oracle experiments to assess performance bottlenecks in the question generation and question answering modules. To scope these evaluations, we only consider input configurations to the rewriting module based on the top two oracle results for 🉐 davinci from Table 2—QA and DQAE. We report these new results in Table 3).

**Question Generation.** First, how much better is QADECONTEXT if we replace generated questions with gold ones? From Table 3, we see a relative lift ranging from **48.2%** to **72.7%** by switching to gold questions (see rows 5 vs 8, 6 vs 9, 7 vs 10). **Question generation is a major source of error.**

**Question Answering.** How much better is *retrieve-then-answer* in QADECONTEXT if we used gold evidence instead of relying on retrieval? Just ablating the *retrieve* step, from Table 3, we only see a modest improvement ranging from **14.2%** to **17.8%** by replacing the retrieved evidence with gold evidence (see rows 3 vs 5, 4 vs 6). **Within-document passage retrieval is not a major source of error.**

How much better is QADECONTEXT if we used gold answers? We ablate the *full document* approach by replacing generated answers with gold

| QG Question | QA Evidence | Answer | DQAE | SARI-add | BERTScore |
|---|---|---|---|---|---|
| *gold* | *gold* | *gold* | ✗✓✓✗ | 0.527 | 0.625 |
| | | | ✓✓✓✓ | 0.427 | 0.580 |
| | | 🔮 gpt-4 | ✗✓✓✗ | 0.274 | 0.541 |
| | | | ✓✓✓✓ | 0.256 | 0.523 |
| | *retrieve* | 🔮 gpt-4 | ✗✓✓✗ | 0.240 | 0.530 |
| | | | ✓✓✓✓ | 0.217 | 0.515 |
| | *full doc* | 🔮 gpt-4 | ✗✓✓✗ | 0.209 | 0.525 |
| 🔮 davinci | *retrieve* | 🔮 gpt-4 | ✗✓✓✗ | 0.139 | 0.481 |
| | | | ✓✓✓✓ | **0.146** | 0.472 |
| | *full doc* | 🔮 gpt-4 | ✗✓✓✗ | 0.141 | 0.495 |

Table 3: Ablating modules in our decontextualization pipeline that affect the input to the final Rewriter module. We ablate (1) source of **question** and (2) use of the full document vs retrieving passages as **evidence**. We investigate including the evidence (E) and source document information (D) in the Rewriter prompt in addition to the questions (Q) and answers (A). Last three rows are fully predictive, while others use gold data.

ones, and see the largest relative improvement: **66.8%** to **92.3%** (see rows 1 vs 3, 2 vs 4). **Question answering is a major source of error.**

**Overall.** While the relative performance improvement from using gold data is large in both the question generation and question answering modules, the absolute values of the scores are quite different. On average, using gold questions provides a 0.080 increase in absolute `SARI-add` (rows 5 vs 8, 6 vs 9, 7 vs 10), while using gold answers provides a 0.212 absolute increase (rows 1 vs 3, 2 vs 4). **We identify question answering as the main performance bottleneck in QADECONTEXT.**

### 6.3 Does QADECONTEXT generalize beyond scientific documents?

We compare our approach to the one used by Choi et al. (2021) by applying our QADECONTEXT strategy to their Wikipedia data. In these experiments, we find that QADECONTEXT performs slightly worse than end-to-end LLM prompting (∼1 percentage point `SARI-add` absolute difference). These results match our intuitions about the QA approach from our formative study (§3.1 and §A.4) in which study participants found that following the QA framework for Wikipedia was cumbersome, was unhelpful, or hindered their ability to perform decontextualization. The results also moti-

vate future work pursuing methods that can adapt to different *document types*, such as Wikipedia or scientific documents, and *user scenarios*, such as snippets being user-facing versus intermediate artifacts in a larger NLP systems. These situations require *personalizing* decontextualizations to diverse information needs.

## 7 Related Work

### 7.1 Decontextualization: Uses and Challenges

Our work is based on Choi et al.'s (2021) seminal work on decontextualization. They show decontextualized snippets can improve passage retrieval. Potluri et al. (2023) show an extract-then-decontextualize approach can help summarization.

Despite its utility, decontextualization remains a challenging task. Eisenstein et al. (2022) noticed similar failures to those we found in §5.1 when dealing with longer input contexts. Beyond models, decontextualization is challenging even for humans. Choi et al. (2021) note issues related to subjectivity resulting in low annotator agreement. Literature in human-computer interaction on the struggles humans have with note-taking (judging what information to include or omit when highlighting) are similar to those we observed in our formative study and data annotation (Chang et al., 2016).

### 7.2 Bridging QA and other NLP Tasks

In this work, we establish a bridge between decontextualization and QA. A similar bridge between QA and discourse analysis has been well-studied in prior NLP literature. In addition to the relevant works discussed in §3.1, we also draw attention to works that incorporate QA to annotate discourse relations, including Ko et al. (2020, 2022); Pyatkin et al. (2020). In particular, Pyatkin et al. (2020) show that complex relations between clauses can be recognized by non-experts using a QA formulation of the task, which is reminiscent of the lowered cognitive load observed during our formative study (§3.1). Beyond discourse analysis, prior work has used QA as an approach to downstream NLP tasks, including elaborative simplification (Wu et al., 2023), identifying points of confusion in summaries (Chang et al., 2023b), evaluating summary faithfulness (Durmus et al., 2020), and paraphrase detection (Brook Weiss et al., 2021).

| | **Question Generation** | |
|---|---|---|
| *Instruction following* | 🌀 davinci might fail to follow requirements specified in the instructions. For example, our prompt explicitly required avoiding questions about figures, which weren't part of the source document. | |
| *Realistic questions* | 🌀 davinci might generate questions that a human wouldn't need to ask as the information is already provided in the snippet. For example, for the snippet *"In addition, our system is independent of any external resources, such as MT systems or dictionaries, as opposed to the work by Kranias and Samiotou (2004).",* 🌀 davinci generated "What kind of external resources were used by Kranias and Samiotou (2004)?" even though the information is already in the snippet (see highlighted text). | |
| *User background* | 🌀 davinci generates questions whose appropriateness depends on user background knowledge. For example, "What is ROUGE score?" is not good question for a user with expertise in summarization. | |
| | **Question Answering** | |
| *Retrieval errors* | 🌀 davinci or 🌀 gpt-4 fails to abstain and hallucinates an answer despite irrelevant retrieved passages. | |
| *Answer errors* | The question is answerable from retrieved context, but 🌀 davinci or 🌀 gpt-4 either unnecessarily abstains or hallucinates a wrong answer. For example, given question: "What does 'each instance' refer to?" and the retrieved passage: *"The main difference was that (Komiya and Okumura, 2011) determined the optimal DA method for each triple of the target word type of WSD, source data, and target data, but this paper determined the method for each instance.",* the model outputs "Each instance refers to each word token of the target data." The correct answer is highlighted. | |
| | **Rewriting** | |
| *Format errors* | 🌀 davinci might fail to enclose snippet edits in brackets. During human evaluation (§5.3), annotators found that 24% of generations had these errors (compared to 5% of gold annotations). | |
| *Missing info* | Overall, annotators found that 45% of decontextualized snippets through QADECONTEXT were still missing relevant information or raised additional questions (compared to 34% for the gold snippets). | |

Table 4: Most common error types at different stages of QADECONTEXT. Question generation and question answering errors identified through qualitative coding of $n = 30$ oracle outputs from §6.2. Rewriting errors identified during human evaluation (§5.3).

## 7.3 QA for User Information Needs

Like in user-facing decontextualization, prior work has used questions to represent follow-up (Meng et al., 2023), curiosity-driven (Ko et al., 2020), or confusion-driven (Chang et al., 2023b) information needs. QA is a well-established interaction paradigm, allowing users to forage for information within documents through the use of natural language (Wang et al., 2022; ter Hoeve et al., 2020; Jahanbakhsh et al., 2022; Fok et al., 2023a).

## 7.4 Prompting and Chaining LLMs

Motivated by recent advancement in instruction tuning of LLMs (Ouyang et al., 2022), several works have proposed techniques to compose LLMs to perform complex tasks (Mialon et al., 2023). These approaches often rely on a pipeline of LLMs to generate to complete a task (Huang et al., 2022; Sun et al., 2023; Khot et al., 2023), while giving a model access to modules with different capabilities (Lu et al., 2023; Paranjape et al., 2023; Schick et al., 2023). While the former is typically seen as an extension of chain-of-thought (Wei et al., 2022), the latter enables flexible "soft interfaces" between models. Our QADECONTEXT strategy relies on the latter and falls naturally from human workflows as found in our formative study.

## 8 Conclusion

In this work, we present a framework and a strategy to perform decontextualization for snippets from scientific documents. We introduce task requirements that extend prior work to handle user-facing scenarios and the handle the challenging nature of scientific text. Motivated by a formative study into how humans perform this task, we propose a QA-based framework for decontextualization that decomposes the task into question generation, answering, and rewriting. We then collect gold decontextualizations and use them to identify how to best provide missing context so that state-of-the-art language models can perform the task. Finally, we implement QADECONTEXT, a simple prompting strategy for decontextualization, though ultimately we find that there is room for improvement on this task, and we point to question generation and answering in these settings as important future directions.

## Limitations

**Automated evaluation metrics may not correlate with human judgment.** In this work, we make extensive use of SARI (Xu et al., 2016) to estimate the effectiveness of our decontextualization pipeline. While Choi et al. (2021) has successfully applied this metric to evaluate decontextualization systems, text simplification metrics present key biases, for example preferring systems that perform fewer modifications (Choshen and Abend, 2018). While this work includes a human evaluation on a subset of our datasets, the majority of experiments rely on aforementioned metrics.

**Collecting and evaluating decontextualizations of scientific snippets is expensive.** The cost of collecting scientific decontextualions limited the baselines we could consider. For example, Choi et al. (2021) approach the decontextualization task by fine-tuning a sequence-to-sequence model. While training such a model on our task would be an interesting baseline to compare to, it is not feasible because collecting enough supervised samples is too costly. In our formative study, we found that it took experienced scientists five times longer to decontextualize snippets from scientific papers compared to ones from Wikipedia. Instead, we are left to compare our method to Choi et al.'s (2021) by running our pipeline in their Wikipedia setting.

The high cost of collecting data in this domain also limited our human evaluation due to the time and expertise required for annotating model generations. For example, a power analysis using $\alpha = 0.05$ and power$= 0.8$, and assuming a true effect size of 5 percentage points absolute difference, estimates that the sample size would be $n = 1211$ judgements per condition for our evaluation in §5.3. Evaluating model generations is difficult for many tasks that require reading large amounts of text or require domain-specific expertise to evaluate. Our work motivates more investment in these areas.

**Closed-source commercial LLMs are more effective than open models.** While we experimented with open models for writing decontextualized snippets (🐪 tülu-30b, 🦙 llama2-chat-70b), results indicate a large gap in performance between their closed-source counterparts, such as 🅰️ claude and 🌀 davinci. Since these systems are not available everywhere and are expensive, their use makes it difficult for other researches to compare with our work, and use our approach.

**Prompting does not guarantee stable output, limiting downstream applicability of the decontextualization approach.** As highlighted in Table 9, all approaches described in this work do not reliably produce outputs that precisely follow the guidelines described in §2. Thus, current systems are likely not suitable to be used in critical applications, and care should be taken when deploying them in user-facing applications.

**Decontextualization is only studied for English and for specific scientific fields.** In this work, we limit the study of decontextualization to natural language processing papers written in English. The reason for this is two-fold: first, most scientific manuscripts are written in English; second, current instruction-tuned LLMs, particularly those that are open, are predominantly monolingual English models.

## Ethical Considerations & Broader Impact

**Reformulation of snippets may inadvertently introduce factual errors or alter claims.** Scientific documents are a mean to disseminate precise and verifiable research findings and observations. Because LLMs are prone to hallucination and may inadvertently modify the semantics of a claim, their use in scientific applications should be carefully scrutinized. Our decontextualization approach is essentially motivated by the need to make snippets portable and understandable away from their source; however, this property makes verification of their content more challenging. While this work does not discuss safeguards to be used to mitigate this risk, these factor must be considered if this research contribution were to be implemented in user facing applications.

**Availability of decontextualization tools may discourage users from seeking original sources.** Because decontextualization systems are not generally available to the public, users today may be more likely to seek the original content of a snippet. Progress in decontexualization systems might change that, as snippets may offer a credible replacement for the full document. We recognize that, while this functionality might offer improvements in scientific workflows, it would also encourage bad scholarly practices. Even more broadly, more general-domain decontextualization systems might lead to users not visiting sources, thus depriving content creators of revenue.

## Acknowledgements

The authors would like to thank Doug Downey, Eunsol Choi, Marti Hearst, Jessy Li, and the Semantic Scholar team at AI2 for useful conversations, participation in user studies, and feedback on our paper draft. We would also like to thank the reviewers for their helpful suggestions and actionable feedback.

## Author Contributions

Benjamin Newman led the project, collected the annotations, implemented all methods, and ran experiments. Luca Soldaini, Arman Cohan, and Kyle Lo were project advisors and provided mentorship. Luca Soldaini also contributed to the code, Kyle Lo conducted the formative study, and the two of them helped with human evaluation. Raymond Fok contributed HCI expertise to the framing of the paper. All authors were involved with writing the paper.

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

## A Appendix

### A.1 Location of necessary context for decontextualizing snippets

See Table 5:

| Source | Location of necessary context | % of snippets |
|---|---|---|
| Wikipedia | Only Title + Context Paragraph | 80-90% |
| Science | Only Context Paragraph | 20.8% |
| | Context Paragraph or Abstract | 51.1% |
| | Cited Paper | 14.8% |

Table 5: Illustration of one reason it is more difficult to decontextualize text from scientific papers compared to Wikipedia. Very often the context needed to decontextualize scientific snippets comes from outside the paragraph with the snippet, including cited papers. This is compared to Choi et al.'s (2021) estimate that the full-text was needed for only 10-20% of the decontextualizations.

### A.2 Types of questions asked about scientific document snippets

We additionally ask annotators to label their questions based on categories we developed while piloting the writing process. We determined that the questions that people ask fall into three categories: (1) Definitions of terms or expansions of acronyms, (2) Coreference resolution, or (3) Simply seeking more context to feel more informed. The annotators' labels are in Table 6:

| | Question Type | | |
|---|---|---|---|
| | **Define** | **Coref** | **More Context** |
| RA | 50 (31%) | 59 (37%) | 52 (32%) |
| CGE | 102 (31%) | 142 (44%) | 82 (25%) |

Table 6: Number of tokens and counts in our dataset, separating out **Research Assistant** (RA) and **Citation Graph Explorer** (CGE) examples.

### A.3 TASP: Selecting important sub-regions of a document when prompting

For models with context windows too small to fit entire papers like 🌀 davinci, we need a condensed representation of the paper to use in prompts. (Choi et al., 2021) find that most of the sentences they decontextualize only require the Title, Section Header of the section the sentence is in, and the paragraph surrounding the snippet. For our snippets from scientific documents, this is likely not sufficient—particularly when paper-specific terms need to be defined. As such, we explore a number of different options:

- **TSP**. **T**itle, **S**ection header, and the **P**aragraph containing the snippet. This is the same condition as (Choi et al., 2021)

- **TASP** and **TAISP**. These add the **A**bstract and **I**ntroduction respectively as both of these contain much of the background context that might need to be incorporated into the snippets.

We found that **TASP** performed best, 0.03 SARI-add points better than **TSP**, and 0.01 points better than **TAISP**. Not including the introductions is potentially helpful because they might include too much distracting information).

### A.4 Additional findings from formative study

In our formative study, we found that stepping through the full framework slows down manual decontextualization. Participants averaged 110 seconds (Wikipedia) and 555 secons (science) per snippet when following **QA** and instead averaged 66 seconds (Wikipedia) and 313 seconds (science) per snippet in the **NoQA** condition. Second, we find no noticeable difference in annotation quality in either setting when operating on Wikipedia snippets. 3 of 5 participants complained that writing down each question and answer was awkward given the simplicity of the task.

### A.5 🌀 davinci vs 🌀 gpt-4 on QA

We compare 🌀 davinci to 🌀 gpt-4 on our question answering step, finding that 🌀 gpt-4 outperforms 🌀 davinci in all cases. The results are visible in Table 7.

### A.6 LLM prompts

The following prompts are for the different stages of the pipeline. They are the prompts for the best-performing models. For prompts for 🔶 claude, 🐪 tülu and 🦙 llama2, please see the github repository linked on the first page.

#### A.6.1 Question Generation

```
The following text is from a scientific paper,
    but might include language that requires
    more context to understand. The language
    might be vague (like "their results") or
    might be too specific (like acronyms or
    jargon). Write questions that ask for
    clarifications. If the language is clear,
    write "No questions.".

Guidelines:
```

| QG Question | QA Evidence | QA Answer | Rewrite DQAE | SARI-add | BERTScore |
|---|---|---|---|---|---|
| *gold* | *gold* | *gold* | ✗✓✓✗ | 0.527 | 0.625 |
| | | | ✓✓✓✓ | 0.427 | 0.580 |
| | | 🌀 davinci | ✗✓✓✗ | 0.233 | 0.536 |
| | | | ✓✓✓✓ | 0.236 | 0.527 |
| | | 🌀 gpt-4 | ✗✓✓✗ | 0.274 | 0.541 |
| | | | ✓✓✓✓ | 0.256 | 0.523 |
| | *retrieve* | 🌀 davinci | ✗✓✓✗ | 0.193 | 0.524 |
| | | | ✓✓✓✓ | 0.190 | 0.518 |
| | | 🌀 gpt-4 | ✗✓✓✗ | 0.240 | 0.530 |
| | | | ✓✓✓✓ | 0.217 | 0.515 |
| | *full doc* | 🌀 gpt-4 | ✗✓✓✗ | 0.209 | 0.525 |
| 🌀 davinci | *retrieve* | 🌀 davinci | ✗✓✓✗ | 0.117 | 0.504 |
| | | | ✓✓✓✓ | 0.140 | 0.483 |
| | | 🌀 gpt-4 | ✗✓✓✗ | 0.139 | 0.481 |
| | | | ✓✓✓✓ | **0.146** | 0.472 |
| | *full doc* | 🌀 gpt-4 | ✗✓✓✗ | 0.141 | 0.495 |

Table 7: Ablating modules in our decontextualization pipeline that affect the input to the final Rewriter module. We ablate (1) source of **question**, (2) use of the full document vs retrieving passages as **evidence**, (3) choice of the QA model for obtaining the **answer**, and (4) the amount of context provided to the rewriter module. Last three rows are fully predictive, while others use gold data. Rewriter module is identical to that from last row of Table 2.

```
* Write the one, two, or three most important
    questions. Do not write unimportant
    questions.
* Do not ask about people or citations.
    Sometimes citations show up as "BIBREF"
* Do not ask questions whose answer is in the
    snippet.
* Do not ask about Tables ("TABREF"), Figures ("
    FIGREF"), Sections ("SECREF") or Formulas ("
    INLINEFORM").

Example:
Snippet: "In spirit, CaRE (Gupta et al., 2019)
    comes closest to our model; however, they do
     not address the problem of type
    compatibility in the link prediction task
    BIBREF3 (See Figure FIGREF2 for details)."
Questions:
- What is "CaRE"?
- What is the authors' approach?
- What is type compatibility?

Snippet: "{{snippet}}"
Questions:
```

### A.6.2 Question Answering

```
Using the given information from the scientific
```

paper, answer the question about "text snippet" below.

```
Information from the paper:
Title: "{{title}}"

Abstract: "{{abstract}}"
Paragraph with potentially helpful information:
    "{{ evidence #1 }}"
Paragraph with potentially helpful information:
    "{{ evidence #2 }}"
Paragraph with potentially helpful information:
    "{{ evidence #3 }}"

Section of the paper the snippet comes from: "{{
    section header}}"
Paragraph with the snippet: "{{paragraph with
    snippet}}"

Text snippet: "{{snippet}}"

Given the above information, please answer the
    following question. Keep your answer concise
     and informative. It should be at most a
    sentence long. If you cannot find the answer
    , then write "No answer.":
Question: {{question}}
```

### A.6.3 Rewriting

```
The following "text snippet" will be quoted in
    an article using the Chicago Manual of Style
    . The following questions were answered
    using information from the paper. Rewrite
    the "text snippet" into quote format by
    adding the answers in-between square
    brackets. Write as if you were an expert
    scientist in the field of natural language
    processing.

Information from the paper:

Question: {{ question #1 }}
Answer: {{ answer #1 }}
Question: {{ question #2 }}
Answer: {{ answer #2 }}
...

Text snippet: "{{sentence}}"

Instructions:

Using the given information, please rewrite the
    text snippet by adding additional
    information into square brackets.
For example: the snippet "Our approach performs
    well" becomes "[REF0's] approach [
    bidirectional language modeling] performs
    well".
For example: the snippet "Our task is MT"
    becomes "[REF0's] task is MT [machine
    translation]."

After adding clarifying information:
* Replace first-person pronouns with a
    placeholder. Replace "we" with "[REF0]" and
    "our" with "[REF0's]".
* Remove discourse markers (like "in conclusion",
     "in this section", "for instance", etc.)
```

* Citations are marked as BIBREF or (Author Name, Year). Keep these the same. Do not add any additional citations.
* Remove any references to Figures ("FIGREF") and Tables ("TABREF")
* Fix the grammar

Please rewrite the snippet according to the instructions and the given information.
Rewrite:

### A.6.4 End-to-End Model (🌀 gpt-4)

system:
  You are a scientist in the field of natural language processing. Using the given information from a scientific paper, rewrite the given text snippet so it stands alone. To do this:
  * Remove discourse markers (like "in conclusion", "in this section", "for instance", etc.)
  * Replace first-person pronouns with placeholders. Replace "we" with "[REF0]" and "our" with "[REF0's]".
  * Remove time-specific words like "current"
  * Make other surface-level changes to fix grammar
  * Resolve any vague or unclear references in the snippet (e.g. "our approach" or "our method")
  * Define any specific terminology or acronyms that other scientists will not be familiar with.
user:
  Using the following scientific paper, rewrite the "text snippet" that follows so it stands alone. The "text snippet" will be quoted in an article using the Chicago Manual of Style. Rewrite the "text snippet" into quote format by adding the answers in-between square brackets.

  Paper:

  {{full_text}}

  Text Snippet: "{{sentence}}"

  Instructions:

  Using only the given information, please rewrite the text snippet into quote format. Specifically add the following clarifying information in square brackets following the Chicago Manual of Style:
  * Resolve any vague or unclear references in the snippet (e.g. "our approach" or "our method"). Put any clarifying text between brackets. For example " Our approach performs well" becomes "[REF0's] approach [bidirectional language modeling] performs well".
  * Define any specific terminology or acronyms that other scientists will not be familiar with. For example "Our task is MT" becomes "[REF0's] task is MT [machine translation]."

* If needed, add additional short clarifications that are necessary for an expert reader to understand the broader context of the quote. Only add up to a single sentence and put the sentence in between square brackets.

  After adding clarifying information:
  * Replace first-person pronouns with a placeholder. Replace "we" with "[REF0]" and "our" with "[REF0's]".
  * Remove discourse markers (like "in conclusion", "in this section", "for instance", etc.)
  * Citations are marked as BIBREF or (Author Name, Year). Keep these the same. Do not add any additional citations.
  * Remove any references to Figures ("FIGREF") and Tables ("TABREF")
  * Fix the grammar

  Reminders:
  * Follow the Chicago Manual of Style for quotes by putting all added text between square brackets.
  * The rewritten snippet is a quote, so the word order should closely match the original snippet's.
  * Ignore irrelevant information.

  Please rewrite this snippet according to the instructions and the given information.
  Text snippet: "{{sentence}}"

### A.6.5 End-to-End Model (🌀 davinci)

  The following "text snippet" will be quoted in an article using the Chicago Manual of Style. Using the given information from scientific paper, rewrite the "text snippet" into quote format by adding in any clarifying information in square brackets. Write as if you were an expert scientist in the field of natural language processing.

Information from the paper:

Title: "{{title}}"

Abstract: "{{abstract}}"
{% if context_section_header %}
Header of section with the snippet: "{{context_section_header}}"
{% endif %}
Paragraph with the snippet: "{{context_paragraph}}"

Text snippet: "{{sentence}}"

Instructions:

Using the given information, please rewrite the text snippet into quote format. Specifically add the following clarifying information in square brackets following the Chicago Manual of Style:
* Resolve any vague or unclear references in the snippet (e.g. "our approach" or "our method

”). Put any clarifying text between brackets
    . For example ”Our approach performs well”
    becomes ”[REF0’s] approach [bidirectional
    language modeling] performs well”.
* Define any specific terminology or acronyms
    that other scientists will not be familiar
    with. For example ”Our task is MT” becomes
    ”[REF0’s] task is MT [machine translation].”
* If needed, add additional short clarifications
    that are necessary for an expert reader to
    understand the broader context of the quote.
    Only add up to a single sentence and put
    the sentence in between square brackets.

After adding clarifying information:
* Replace first-person pronouns with a
    placeholder. Replace ”we” with ”[REF0]” and
    ”our” with ”[REF0’s]”.
* Remove discourse markers (like ”in conclusion”,
    ”in this section”, ”for instance”, etc.)
* Citations are marked as BIBREF or (Author Name,
    Year). Keep these the same. Do not add any
    additional citations.
* Remove any references to Figures (”FIGREF”)
    and Tables (”TABREF”)
* Fix the grammar

Reminders:
* Follow the Chicago Manual of Style for quotes
    by putting all added text between square
    brackets.
* The rewritten snippet is a quote, so the word
    order should closely match the original
    snippet’s.

Please rewrite this snippet according to the
    instructions and the given information.
Text snippet: ”{{sentence}}”
Rewrite:

## A.7 Sample QA Pairs

| | |
|---|---|
| *Title:* *"DOLORES: Deep Contextualized Knowledge Graph Embeddings"* | |
| *User query:* *"Is fine-tuning required to incorporate these embeddings into existing models?"* | |
| *Original Snippet* | The only requirement is that the model accepts as input, an embedding layer (for entities and relations). If a model fulfills this requirement (which a large number of neural models on knowledge graphs do), we can just use Dolores embeddings as a drop-in replacement. We just initialize the corresponding embedding layer with Dolores embeddings. |
| *QA-Pairs* | Question: "What is an embedding layer?" |
| | Answer: "An embedding layer is a layer in a neural network model that accepts as input representations of entities and relations in the form of embeddings." |
| | Question: "What are Dolores embeddings?" |
| | Answer: "Dolores embeddings are deep representations of entities and relations in knowledge graphs, learned using Bi-Directional LSTMs from entity-relation chains." |
| | Question: "How do we initialize the corresponding embedding layer?" |
| | Answer: "We initialize the corresponding embedding layer with Dolores embeddings." |
| Decontextualized Snippet (SARI-add: 0.176) | [REF0's] model accepts input representations for entities and relations in the form of dense continuous vector embeddings [i.e., an embedding layer]. Dolores embeddings, which are deep contextualized knowledge graph embeddings learned using a deep neural sequential model, can be used as a drop-in replacement for the embedding layer in existing knowledge graph prediction models. To initialize the corresponding embedding layer, [REF0] simply uses Dolores embeddings. |
| *Citing paper:* *"Extracting Social Networks from Literary Text with Word Embedding Tools"*(Wohlgenannt et al., 2016) | |
| *Cited paper:* *"Extracting Social Networks from Literary Fiction"*(Elson et al., 2010) | |
| *Original Snippet* | In contrast to our work, (Elson et al., 2010) are solely focus on length and number of dialogues between persons to measure relatedness, whereas our approach looks at general co-occurrence or similarity as measured by LT tools which use word embeddings. |
| *QA-Pairs* | Question: "What are LT tools?" |
| | Answer: "LT tools are language technology tools that use word embeddings for measuring similarity and co-occurrence in text." |
| | Question: "How do LT tools measure co-occurrence or similarity?" |
| | Answer: "LT tools measure co-occurrence or similarity using word embeddings." |
| | Question: "What are word embeddings?" |
| | Answer: "Word embeddings are numerical representations of words in a multi-dimensional space, which capture semantic and syntactic information about the words and their relationships with one another." |
| Decontextualized Snippet (SARI-add: 0.148) | [REF0's] approach looks at general co-occurrence or similarity as measured by LT [language technology] tools, which use word embeddings [language modeling techniques that transform the vocabulary of an input corpus into a continuous and low-dimensional vector representation, capturing semantic and contextual information of words]. |

Table 8: Two examples of the outputs of the different stages our best decontextualization pipeline. The questions, answers, and decontextualized snippet are all model generated. The first example is from the QASPER dataset (Dasigi et al., 2021); the snippet is an evidence passage containing the answer the user question. The second is a text span extracted from Wohlgenannt et al. (2016) citing Elson et al. (2010). Note that the questions are not all natural and are sometimes redundant, but the information they query is only included once in the decontextualized snippet.

| | |
|---|---|
| *Title:* "DOLORES: Deep Contextualized Knowledge Graph Embeddings" | |
| *User query:* "Is fine-tuning required to incorporate these embeddings into existing models?" | |
| *Original Snippet* | The only requirement is that the model accepts as input, an embedding layer (for entities and relations). If a model fulfills this requirement (which a large number of neural models on knowledge graphs do), we can just use Dolores embeddings as a drop-in replacement. We just initialize the corresponding embedding layer with Dolores embeddings. |
| *Gold* | The only requirement is that the model accepts as input an embedding layer (for entities and relations). If a model fulfills this requirement (which a large number of neural models on knowledge graphs do), the authors can just use Dolores embeddings [which are created using connections between language models and random walks on knowledge graphs] as a drop-in replacement. The authors just need to initialize the corresponding embedding layer with Dolores embeddings. |
| Best Oracle Pipeline (SARI-add: 0.522) | The only requirement is that the model accepts as input, an embedding layer [for entities and relations]. If a model fulfills this requirement (which a large number of neural models on knowledge graphs do), [REF0] can just use Dolores embeddings as a drop-in replacement. [REF0] just has to initialize the corresponding embedding layer with Dolores embeddings, which can be achieved by creating connections between language models and random walks on knowledge graphs. |
| Best Pipeline (SARI-add: 0.176) | [REF0's] model accepts input representations for entities and relations in the form of dense continuous vector embeddings [i.e., an embedding layer]. Dolores embeddings, which are deep contextualized knowledge graph embeddings learned using a deep neural sequential model, can be used as a drop-in replacement for the embedding layer in existing knowledge graph prediction models. To initialize the corresponding embedding layer, [REF0] simply uses Dolores embeddings. |
| *Citing paper:* "Extracting Social Networks from Literary Text with Word Embedding Tools"(Wohlgenannt et al., 2016) | |
| *Cited paper:* "Extracting Social Networks from Literary Fiction"(Elson et al., 2010) | |
| *Original Snippet* | In contrast to our work, (Elson et al., 2010) are solely focus on length and number of dialogues between persons to measure relatedness, whereas our approach looks at general co-occurrence or similarity as measured by LT tools which use word embeddings. |
| *Gold* | In contrast to the authors' work [based on co-occurence statistics and cosine similarity], (Elson et al., 2010) focus solely on length and number of dialogues between persons to measure relatedness, whereas the authors' approach looks at general co-occurrence or similarity as measured by [Language Technology] (LT) tools which use word embeddings. |
| Best Oracle Pipeline (SARI-add: 0.5) | [REF0's] approach [experimentation based on co-occurence statistics and cosine similarity] looks at general co-occurrence or similarity as measured by LT tools [state of the art word embedding tools] which use word embeddings. |
| Best Pipeline (SARI-add: 0.148) | [REF0's] approach looks at general co-occurrence or similarity as measured by LT [language technology] tools, which use word embeddings [language modeling techniques that transform the vocabulary of an input corpus into a continuous and low-dimensional vector representation, capturing semantic and contextual information of words]. |

Table 9: Two examples of our decontextualization pipeline compared with gold annotations and end-to-end output from GPT-3. The first example is from the QASPER dataset (Dasigi et al., 2021); the snippet is an evidence passage containing the answer the user question. The second is a text span extracted from Wohlgenannt et al. (2016) citing Elson et al. (2010) Together, they demonstrate how an effective decontextualization system can improve consumption of text outside the originating document. Text in blue has been added by the systems.