# OpenReview forum: "A Question Answering Framework for Decontextualizing User-facing Snippets from Scientific Documents"
_EMNLP/2023/Conference — EMNLP 2023 Main_

### Official Review · Reviewer_BSQ8 · 2023-07-23

**Soundness:** 3

**Ethical Concerns:**

Yes

**Excitement:**

3: Ambivalent: It has merits (e.g., it reports state-of-the-art results, the idea is nice), but there are key weaknesses (e.g., it describes incremental work), and it can significantly benefit from another round of revision. However, I won't object to accepting it if my co-reviewers champion it.

**Justification For Ethical Concerns:**

The same listed on page 9. In particular, any kind of false/error in the inferences might lead to some legal/ethical issues such as attributing false claims to some authors, or imprecise conclusions that can lead other studies to something wrong.


**Paper Topic And Main Contributions:**

this paper proposes a strategy to decontextualize scientific documents. In so doing, their framework divide the challenge into question generation, answering and rewriting. The authors also designed a gold standard for their study.



**Questions For The Authors:**



* Authors need to put substantial efforts in rewriting the paper in order to enhance readability. I ended up reading the abstract very confused, for example. There are also some grammar error across the manuscript.

a) "Many real-world applications (e.g., note taking, search) require extracting a sentence or paragraph from a document and showing that snippet to a human outside of the source document." This phrase needs a little of rephrasing as a means of enhancing readability.


b) "Yet, snippets divorced from their origin are often missing context and can be in comprehensible when presented to users" Same here, maybe a better crafting is as follows: "However, these snippets can be hard to understand [for a human reader] when presented outside from their context"


c) Many sentences are very long thus take some extra effort to read and understand. Consider splitting and linking properly.

d) "Second, we propose a framework that decomposes the task into question generation, question answering, and rewriting stages."
After reading this sentence, I ask myself "what are the authors decomposing?"


e) "Using this framework, we collect gold decontextualizations from experienced readers of scientific articles and conduct a range of experiments across state-of-the-art commercial and open-source language models to identify how best to provide missing-but-relevant in formation to models for our task." This sentence needs rephrasing and splitting.

f) In the definition (lines 115-119), it is explicitly defined "c".
It looks like that snippet is wrongly "decontextualized".

g) "Finally, to handle cross-document references, we assume in the QA application setting that a user would have access to basic information about the current document of interest but no knowledge  about any referenced documents that may appear in the snippet text." This is not realistic, especially if a ground-breaking work was cited like figure 1.

h) Some examples of each of the three steps should be added to figure 2 for the sake of clarity

i) "A clarification matches if at least 75% of the added tokens in the prediction match one of the targets" .. What about highly frequent terms such as play, experiment, and results. Please answer as it relates to the CLF metrics.

j) "Surprisingly, the QA module of the pipeline is the weakest."
Why was it a surprise? I think it is the hardest component and its gold standard provides lot of new pieces of information. I would say "as a conclusion" instead of "Surprisingly"















**Reasons To Accept:**


* The experimental section is nice and clean. They provide nice features in their analysis.

* The problem is challenging and interesting.

*The paper can be understood by a wider audience.

* Results provides insight into how to move forward in this task.








**Reasons To Reject:**

* Error analysis and significance tests are missing.

* The studied problem might be of interest of a narrow audience.

* The English needs improvements.



**Reproducibility:**

3: Could reproduce the results with some difficulty. The settings of parameters are underspecified or subjectively determined; the training/evaluation data are not widely available.

**Reviewer Confidence:**

3: Pretty sure, but there's a chance I missed something. Although I have a good feel for this area in general, I did not carefully check the paper's details, e.g., the math, experimental design, or novelty.

---

> ### Author Rebuttal · Authors · 2023-08-29
>
> The authors would like to thank Reviewer BSQ8 for their thorough and careful review. We appreciate that the reviewer believes the experiments are “clean” and that “the problem is challenging and interesting”.
>
> ****
>
> **_Concern 1:_**  Lack of significance testing reported for human evaluation.
>
> **_Response 1:_** We have performed these tests on an expanded human evaluation. To avoid too much overlap in our rebuttals, please see our **Response 3 to Reviewer uGwc** for details.
>
> ****
>
> **_Concern 2:_**  Concerns about our clarity of writing.
>
> **_Response 2:_** We appreciate the suggestions to improve our writing clarity. **We edited the sentences surfaced by the Reviewer,** as well as others that are long and difficult to read. (Addressing the reviewer’s questions “a-f” and “j”)
>
> Additionally, the reviewer suggests adding examples of the outputs of each step of the pipeline to Figure 2 (h). We agree that adding examples will clarify the paper. **We will use our additional camera ready page to add a new Table with an example generation from each step to the main paper.** This Table will also serve for illustrating error analysis (see Reviewer’s concern #4 below). Additionally, **we will add more to the Appendix of the paper**.
>
> ****
>
> **_Concern 3:_** Reviewer BSQ8 asks how common words might skew the CLF metric.
>
> **_Response 3:_** Like many n-gram based scores (such as SARI), we do not explicitly weigh the token types by frequency when calculating CLF. In preliminary evaluations, we found that removing stop words were sufficient for using CLF to distinguish between systems.
>
> ****
>
> **_Concern 4:_** More In-depth error analysis for the pipeline.
>
> **_Response 4:_** We have performed this analysis and agree the results would be informative for readers. We plan to include it in a new Table in the main body as part of our additional camera ready page. We iterated on model prompts to minimize these errors, but still some occurred. We describe our findings for each pipeline step below:
>
> **Question Generation:**
>
> - Asking about Citations
>
>   - E.g. Snippet: “While Shen et al. (2019) makes RNN decoder as a MoE…”
>   - Question: “Who are Shen et al. (2019)?"
>
> - Asking questions that are answered in the text.
>
>   - E.g. Snippet: “In addition, our system is independent of any external resources, such as MT systems or dictionaries, as opposed to the work by Kranias and Samiotou (2004).”
>   - Question: “​​What kind of external resources were used by Kranias and Samiotou (2004)?”
>
> - Asking questions that an expert likely wouldn’t ask:
>
>   - Snippet: “Concretely, we apply the … model trained on the full Reddit dataset to millions of new unannotated posts, labeling these posts with a probability of dogmatism according to the classifier.”
>    - Question: "How is the classifier labeling posts with a probability of dogmatism?"
>
>
> **Question Answering**
>
> There were two main sources of errors in the QA portion of the pipeline that can be broken down into two categories:
>
> - Retrieval Errors: The answer **was not** in the retrieved context, so the model just made something up rather than abstaining.
> - QA Errors: The answer **was** in the retrieved context, but the model still answered the question incorrectly or unnecessarily abstained.
>
>
> **Rewriting:**
>
> - In fact, we have some of this analysis already in Table 5, when we report on human evaluation. To make this more prominent, we will move out this content from Table 5, leaving it to focus solely on the accept/reject judgment component of the human evaluation. The error analysis will go into the new table with the other pipeline errors.
> - Our findings are still what we report in our submission: Most rewriting errors are **formatting** (usually not putting the added information between brackets, which is a requirement of user-facing snippets) or **missing information** (the system missing important information that is in the gold decontextualization).
>
> ****
>
> Thank you again for your close reading and thoughtful reviews of our paper. We hope that the changes we have made sufficiently address the reviewers’ comments, that the reviewer increases their scores.

---

### Official Review · Reviewer_uGwc · 2023-08-02

**Typos Grammar Style And Presentation Improvements:** The overall presentation of the paper…
**Soundness:** 4

**Excitement:**

4: Strong: This paper deepens the understanding of some phenomenon or lowers the barriers to an existing research direction.

**Paper Topic And Main Contributions:**

The paper introduces a question-answering based framework for de-contextualization -- the task of adding necessary context to a sentence from the document it appears in to make the sentence standalone. The proposed framework features a three-stage pipeline -- Given a sentence, the framework (1) generates clarifying questions, (2) Extract answers to the questions from other sentences in a document, and (3) De-contextualize and rewrite the sentence based on the answers.

The paper studies the problem in the domain of scientific documents, and conducts a small human study to analyze the needs and challenges of decontextualization from a user perspective. The proposed system uses different combinations of LLMs for the three parts of the pipeline. The authors provide evaluation and ablation for the overall system.

**Questions For The Authors:**

I'm curious how InstructGPT (davinci) vs. GPT4 looks like in the end2end setting? From the human evaluations, it seems that GPT-4 is selected the best end2end system, but it wasn't clear from Table 2.

On a related note, it seems that CLF scores don't have a strong correlation with other metrics. It might be worth adding a bit more discussion and analysis to help explain why that is.

**Reasons To Accept:**

The paper presents an in-depth evaluation and ablation studies for using LLMs to do QA-QG-rewrite based decontextualization. The authors show the performance gap between using gold annotated question, answer, evidence vs. the model generated ones, where the later setting is more realistic. The author shows that a QA-QG-rewrite pipeline system can perform better than end-to-end system in 0-shot settings. The conceptual idea of the method is very interesting.

**Reasons To Reject:**

Since decontextualization is an established task, IMO the evaluation in the paper could be greatly strengthened by comparing to an established baseline, e.g. a finetuned seq2seq system with dataset from Choi et. al. 2021. This will --
1. Help explain the conceptual differences between the decontextualization task for Wikipedia vs. "user-facing" scenarios, e.g. scientific documents, in a more quantitative way.
2. Help readers/users understand the capability of using LLM for decontextualization vs. supervised baselines.

IMO there are two ways to go about this, either --
1. Test the QA-QG-rewrite pipeline on the Choi et. al. 2021 test set (i.e. Wikipedia sentences), and compare to the numbers there, or
2. Train a supervised baseline from Choi et. al. 2021, and test on your data.

Apart from baseline choices, here are some additional comments on evaluation settings
- It seems that all LLM prompts are used in zero-shot setting without providing in-context examples. I understand that this is needed for fitting an entire paper in the context window of a single prompt forward. But this seems really disadvantageous for the end2end model, as it's a relatively complex and procedural task. IMO using even a single demonstration could make the performance much better. It's worth dedicating some discussion + analysis to compare the performance in zero- vs. few-shot settings (even if this can only be done on a subset of the dataset)
-  Since the human evaluation is done only on 30 snippets, it's worth checking if the improvements are statistically significant.

**Reproducibility:**

4: Could mostly reproduce the results, but there may be some variation because of sample variance or minor variations in their interpretation of the protocol or method.

**Reviewer Confidence:**

4: Quite sure. I tried to check the important points carefully. It's unlikely, though conceivable, that I missed something that should affect my ratings.

---

> ### Author Rebuttal · Authors · 2023-08-29
>
> We are grateful to Reviewer uGwc for their insightful questions and suggestions. We are glad that the reviewer found our “in-depth” evaluation of our proposed decontextualization method “conceptually very interesting”. Below, we summarize and address Reviewer uGwc’s questions and concerns:
>
> ****
>
> **_Concern 1.1_**_:_ Additional baselines would make our work more comparable to prior work.
>
> Option 1: Train a supervised model on our data like Choi et al., 2021 did on theirs.
>
> **_Response 1.1:_** We carefully considered our evaluation approach and experimental setting, and we appreciate the opportunity to dive deeper into our decision-making.
>
> We did not pursue a supervised baseline because **collecting large-scale data for scientific decontextualization is too costly**. During our formative study investigating the difference between Choi, et. al. (2021)’s and our scientific decontextualization setting, we found that our setting takes **five times longer** to annotate compared to Wikipedia snippets (details in Appendix A.4). Annotation time and difficulty in recruiting and fairly compensating qualified experts prevented us from collecting enough data for supervised finetuning.
>
> Recent decontextualization work also prefers prompting over fine-tuning. For example, Eisenstein et al., 2022 (cited in the text) find that a seq2seq model fine-tuned on \~11k examples performs worse at decontextualizing Wikipedia sentences than a prompted LLM (PaLM). Overall, we recognize the limitation of not having a supervised baseline, but we hope the Reviewer finds this is a reasonable tradeoff given the other experiments we’ve invested in. We will make this clearer in our Limitations section.
>
> ****
>
> **_Concern 1.2:_** Additional baselines would make our work more comparable to prior work.
>
> Option 2: Test our pipeline on Wikipedia data from Choi et al., 2021.
>
> **_Response 1.2:_** This option is feasible, and we agree such an experiment would help the reader better understand our setting vs the simpler Wikipedia setting in a quantitative manner. In such experiments, we observe that **our QA-based approach performs slightly worse than end-to-end LLM prompting on Wikipedia** (\~1 percentage point SARI-add absolute difference). This matches our intuitions about the QA approach from our formative study (Section 3 & Appendix A.4) in which **our study participants found that following the QA framework for Wikipedia was cumbersome** and either was unimportant or hindered their ability to perform decontextualization. **Our formative study and experiments illustrate that the QA framework’s benefit over end-to-end baselines is a consequence of our more challenging setting**.
>
> Our quantitative analysis in Table 6 of Appendix 1a also enriches the comparison between the Wikipedia and scientific settings. There we analyzed the question: _how much full text is needed to perform our task?_ We found that **only 20% of our snippets can be decontextualized using only the surrounding paragraph; in contrast, Choi et al. report being able to decontextualize 80-90% using just the surrounding paragraph**. We hope together these results paint a better quantitative picture of the Scientific setting vs the Wikipedia setting. We thank the Reviewer for this suggestion and will include it in the camera-ready using our additional page.
>
> ***
>
> **_Concern 2:_** Because we need to fit the entire paper into the context window, we can’t provide in-context examples, which could disadvantage the model.
>
> **_Response 2:_** We agree with the reviewer that not using in-context examples could potentially disadvantage the end-to-end models. In fact, **we do include examples as part of model instructions in the end-to-end prompting baselines**. As one example, in our end-to-end prompt, we include the following instructions:
>
> _Resolve any vague or unclear references in the snippet (e.g. "our approach" or "our method"). Put any clarifying text between brackets. For example "Our approach performs well" becomes "\[REF0's] approach \[bidirectional language modeling] performs well"._
>
> This question made us realize that the prompts in the Appendix A.5 were from an older version of the system, before we iterated on the prompts. Thank you for pointing this out - it has been corrected for the camera-ready version. The full prompt is available at our anonymous code link in the paper, at the path `configs/templates/full_paper_chat`.
>
> ****
>
> **_Concern 3:_**  Lack of significance testing reported for human evaluation.
>
> **_Response 3:_** To address this feedback, **we have performed more human evaluation.**
>
> - First, we performed double-annotation on the previous human annotation data; based on these, we estimated binary agreement between two annotators of **0.7**. We believe this is reasonably high given our task is quite challenging and subjective; Choi et al reported agreement for Wikipedia snippets around 0.8 and also discussed inherent subjectivity of this task.
> - Next, we had both annotators annotate **60 new generations using GPT3-davinci** for end-to-end instead of GPT4, based on the Reviewer’s suggestion. This was an oversight on our part, and we should have used the one with the higher SARI score. Thanks to the Reviewer for catching this.
>
> We find the **same human eval result (Table 5) holds from before: Gold > Pipeline > End-to-End.**  We also find the Pipeline and End-to-End acceptability scores are still close (**5-7 percentage points difference**).
>
> We conducted a **two-sample Binomial test** using just the new **n=60** judgements as well as aggregating all judgments **(n=90)**. In both cases, the difference between Pipeline and End-to-End was **not statistically significant** (p=0.57 under n=60; p=0.45 under n=90). We conducted a power analysis using $\alpha$=0.05 and power=0.80, and assuming a true effect size of 5 percentage point absolute difference, **estimated sample size would’ve been n=1211 judgments per condition**. Unfortunately, this would’ve been prohibitively costly for us to collect.
>
> In conclusion, we believe it’s important to be transparent with our findings. **We will add the following to our paper**:
>
> - The additional human evaluation results (increased sample size, agreement, and switch to davinci end-to-end) to Table 5
> - The statistical analysis in Section 5.5 when interpreting results
> - A more clear discussion of the sample size issue in the Limitations
> - Add discussion about scalable human evaluation as an important future challenge that NLP research must tackle (similar scalable issues for human evaluation are being seen in other generative NLP areas, like summarization). Our task can hopefully be yet another motivating example for more investment in this important area.
>
> ****
>
> **_Concern 4:_** Reviewer uGwc wonders why CLF seems not to be correlated with the other metrics.
>
> **_Response 4:_** We would not necessarily expect the CLF metric to be the same as the others. CLF measures the precision, recall, and F1 of the added clarifications, rather than added tokens. For example, if a decontextualization has two clarifications, one short (e.g. expanding an acronym) and one long (e.g. describing the method of a cited paper) and the prediction includes tokens in the longer one, but misses the shorter one, the SARI score would be high, but the CLF score would be low. Cases like these lead to the discrepancies. We will add this clarification to our discussion about CLF.
>
> ****
>
> The authors would like to thank Reviewer uGwc for their detailed feedback, questions, and corrections. We hope these updates address the main concerns with the paper, and if so, we hope the reviewer updates their scores.

---

### Official Review · Reviewer_ELeZ · 2023-08-03

**Soundness:** 5

**Excitement:**

5: Transformative: This paper is likely to change its subfield or computational linguistics broadly. It should be considered for a best paper award. This paper changes the current understanding of some phenomenon, shows a widely held practice to be erroneous in someway, enables a promising direction of research for a (broad or narrow) topic, or creates an exciting new technique.

**Paper Topic And Main Contributions:**

The article tackles decontextualization, a comparatively new field of research whose success is currently closely tied to the advances of LLMs. It describes and operationalizes the task of rewriting snippets to incorporate information from their originating contexts. It also evaluates several methods on a specially created dataset.

**Questions For The Authors:**

Question A: You do not seem to mention using Upwork and Prolific in the limitations yet there are potential risks, mostly of technical (malpractice by workers, use of LLMs to perform the tasks, English proficiency) and ethical (e.g. worker farms scattered in low-income countries) nature. Did you account for such shortcomings?

Question B: Between the time of submission and the present state of the art open models have made significant progress so that your assertion line 626 could be modified ("Closed-source commercial LLMs make for effective pipelines; open models do not"). Can you comment on this point? Did you measure progress by open models other than tülu?

**Reasons To Accept:**

The paper is scientifically sound, the state of the art comprehensive, the goals are clearly stated and the evaluation is convincing.

There are a number of open questions regarding the definition and the evaluation of the task but as this field of research is relatively new and as the authors openly discuss the matter it is yet another reason to accept the submission.

**Reasons To Reject:**

As the dataset described in 4.3 is fairly small and the methods used are quite complex there might be unknown biases affecting the evaluation. The use of commercial models is subject to change over time which make the evaluation impossible to reproduce or results difficult to interpret.

**Reproducibility:**

4: Could mostly reproduce the results, but there may be some variation because of sample variance or minor variations in their interpretation of the protocol or method.

**Reviewer Confidence:**

3: Pretty sure, but there's a chance I missed something. Although I have a good feel for this area in general, I did not carefully check the paper's details, e.g., the math, experimental design, or novelty.

**Typos Grammar Style And Presentation Improvements:**

Line 410: For the sake of clarity you could define the SARI method or at least reveal what's behind the acronym.

---

> ### Author Rebuttal · Authors · 2023-08-29
>
> We thank reviewer ELeZ for recognizing our work on this “comparatively new field of research” as “sound” and “comprehensive”. We agree that our “open discussion” of task definition and evaluation is important, especially for engaging with future researchers to explore relatively nascent areas of NLP such as ours. Below, we summarize and address reviewer ELeZ’s questions and concerns:
>
> ***Concern 1:*** Use of commercial systems in evaluation
>
> ***Response 1***: We agree that it is difficult to interpret and reproduce evaluation results from commercial systems, especially as they change over time. Commercial systems are used by many people, which is why we felt it was important to include them in our evaluation. **We demonstrate that our setting is challenging even for these commercial systems, which helps raise awareness and recalibrate public perception around the strengths and limitations of these systems.** This recalibration is especially important as many of the scientific applications cited in our introduction are built using commercial models. Testing on multiple commercial systems also gives us confidence that our protocol will work on open models as their performance improves.
>
> Nevertheless, for scientific reproducibility and our findings’ utility for the broader research community, we agree with the importance of evaluating using open models as well. This is why we also **performed evaluation using the open model Tülu** as it was state-of-the-art on a wide range of tasks at the time of submission. After our paper submission, we also **evaluated the newly released Llama 2 model** (70B chat) and found it to outperform Tülu, but still underperform commercial systems. Specifically, compared to the “best setting” results in Table 1 (bottom row), the SARI-add scores for Llama 2 are 0.342 compared to Tülu (0.252). Furthermore, Llama 2’s increased context length gives us the ability to add results that previously weren’t feasible for Tülu. For example, including the full document + QA pairs + evidence results in a 0.205 SARI-add score compared to 0.378 (Claude) or 0.427 (davinci). We are excited to add the full set of Llama 2 results to Table 1 to the camera-ready, as they indicate rapid advancement in available open models, while still leaving much room for improvement compared to commercial systems.
>
> **_Concern 2_**: How did we select annotators who created our dataset?
>
> **_Response 2_**: We agree that, as with any data collection, one must safeguard against worker exploitation and low-quality annotations. **We took care to avoid potential technical and ethical malpractice and ensure high-quality annotations**. First, **the size of our dataset allowed us to hand-verify every example we collected. This verification took the authors over 20 hrs to conduct.** To incentivize workers, we also paid a reasonably high rate for annotator work:
>
> - On Upwork, we hired four contractors and paid them each $20/hr USD to create the questions. These annotators were first given a set of pilot annotations after which we clarified their questions about the annotation protocol.
> - On Prolific, we avoided the issue of “worker farms” by limiting participants to participants in the USA or the UK and paying them $17/hr USD.
>
> We will make this information clearer in Section 4.2.
>
> We’d again like to thank Reviewer ELeZ for their thoughtful questions; the constructive feedback has helped us improve our manuscript. We hope our responses have increased their confidence in our work.

---

### Meta-Review · Area_Chair_VbKn · 2023-09-17

**Recommendation:** 4

**Metareview:**

This paper tackles decontextualization through the lens of question generation and question answering. Decontextualization is a relatively new but important task; the goal is to rewrite an extractive (short) text snippet such that it "stands alone". This work builds on prior work (a seq2seq system from Choi et al), and introduces a 3-step process with question generation, question answering, and rewriting. This framework, inspired by prior work in QUD, is compelling and useful in this task. This work involves an annotated dataset and sound evaluation of LLMs.

The reviewers pointed out several issues to be addressed; notably please include the answers in the author response to reviewer questions, such as new results re LLAMA2, additional human evaluation, additional error analysis, and other clarifications.

---

### Decision · Program_Chairs · 2023-10-07

**Decision:**

Accept-Main

**Comment:**

This paper tackles decontextualization through the lens of question generation and question answering. Decontextualization is a relatively new but important task; the goal is to rewrite an extractive (short) text snippet such that it "stands alone". This work builds on prior work (a seq2seq system from Choi et al), and introduces a 3-step process with question generation, question answering, and rewriting. This framework, inspired by prior work in QUD, is compelling and useful in this task. This work involves an annotated dataset and sound evaluation of LLMs.

The reviewers pointed out several issues to be addressed; notably please include the answers in the author response to reviewer questions, such as new results re LLAMA2, additional human evaluation, additional error analysis, and other clarifications.